# Effect of the Mixed Inoculation of Lactic Acid Bacteria and Non-*Saccharomyces* on the Quality and Flavor Enhancement of Fermented Mango Juice

Qiuping Zhong [1,2,*,†], Ruixin Chen [1,2,†], Ming Zhang [1,2], Wenxue Chen [1,2], Haiming Chen [1,2] and Weijun Chen [1,2]

1    School of Food Science and Engineering, Hainan University, Haikou 570228, China; hainufood97@163.com (R.C.); zhangming-1223@163.com (M.Z.); chwx@hainanu.edu.cn (W.C.); hmchen168@126.com (H.C.); chenwj@hainanu.edu.cn (W.C.)
2    Key Laboratory of Food Nutrition and Functional Food of Hainan Province, Haikou 570228, China
*    Correspondence: 990511@hainanu.edu.cn; Tel.: +86-0898-6619-3581
†    These authors contributed equally to this work.

**Abstract:** Mango juice (MJ) was co-inoculated with *Lactobacillus plantarum + Rhodotorula glutinis* or *Metschnikowia pulcherrima* (LP + RG or LP + MP, respectively) and *Lactobacillus casei + Rhodotorula glutinis* or *Metschnikowia pulcherrima* (LC + RG or LC + MP, respectively) to evaluate their effect on the physicochemical characteristics, antioxidant capacity, and aroma compounds of MJ after 72 h of fermentation at 28 °C. Results indicated that among the fermented MJ, that which was fermented with LC + RG yielded the highest content of total acid (15.05 g/L). The pH values of MJ fermented with LC + MP, LC + RG, LP + RG, and LP + MP were 3.36, 3.33, 3.26, and 3.19, respectively, and were lower than that of CK (4.79). The juice fermented with LP + MP culture had the lowest sugar content (73.52 g/L), and those fermented with LP + RG and LP + MP had higher total phenol contents and stronger DPPH radical scavenging activity, ABTS radical scavenging activity, iron-reducing antioxidant capacity, and copper reducing antioxidant capacity than the others. Carotenoids in MJ had varying degrees of degradation after mixed fermentation by using all four combinations. Volatile compounds revealed that the co-fermentation of LP + RG produced increased norisoprenoid aroma compounds. The mixed co-inoculation method is a strategy to contemplate for MJ fermentation, but the modalities of inoculation need further investigation. Success depends on the suitable combination of non-*Saccharomyces* and lactic acid bacteria and consideration of strain variation.

**Keywords:** mango juice; mixed fermentation; lactic acid bacteria; non-*Saccharomyces*; aroma; norisoprenoids

## 1. Introduction

Mango (*Mangifera indica* L.) is the most important commercial fruit due to its intense aroma, delicious flavor, and high nutritional value [1]. Among the various processed mango products, mango juice (MJ) is the most favored because of its pleasing organoleptic qualities and rich nutritional value [2].

Flavor plays a crucial role in the quality and acceptability of MJ products. MJ is suitable for use in the production of probiotic beverages via biofermentation. Reddy et al. [3] used lactic acid bacteria (LAB) to ferment MJ and found that *Lactobacillus plantarum* (LP) utilizes the sugar and decreases the pH quickly. The viability of cells is maintained at $10^7$ CFU/mL throughout the storage period. Coulibaly et al. [4] used non-*Saccharomyces*-fermented MJ and observed reduced sugar content, increased contribution of flavor and aroma to the juice, and suppression of undesirable microorganisms.

The FMJ produced by monocultures of LAB often lacks the complexity of flavor and character, but monocultures of non-*Saccharomyces* are sometimes related to low acidity, low

fermentation power, and poor taste [5]. However, in controlled mixed-culture fermentation, these disadvantages of non-*Saccharomyces* and LAB may not be expressed, and the advantages of having both kinds of microorganisms may be properly manifested.

Recently, more studies highlighted the positive roles of non-*Saccharomyces* yeast or LAB in controlled mixed-culture fermentation. The simultaneous and successive inoculations of *Rhodotorula mucilaginosa* and *Saccharomyces cerevisiae* to ferment ecologically dry white wines revealed that mixed fermentation improves the composition of varietal and fermentation aromatic compounds and enhances citrus, sweet fruit, tart fruit, berry, and floral aromatic traits [6]. Sadineni et al. [7] found that *Torulaspora delbrueckii/Metschnikowia pulcherrima* (MP) fermentations mixed with *S. cerevisiae* may be important to the mango winemaking industry. Mixed cultures can be used to improve product quality, enhance aroma complexity, reduce volatile acids, and modify some undesired parameters of the final wine. Hu et al. [8] also pointed out that the selection of non-*Saccharomyces* yeast strains for co-fermentation with *S. cerevisiae* is a promising method for improving the organoleptic quality of fruit wines. Jin et al. [9] used LP and *S. cerevisiae* DV10 for the mono- and co-culture fermentation of mango puree and found that the mixed co-culture reduces terpenes and produces alcohols and esters. Co-cultures obtain the highest total phenolic content (TPC) and show the strongest antioxidant activity.

Studies pointed out that many of the important fruit aroma volatiles come from the degradation of carotenoid pigments [10]. Norisoprenoids are compounds with specific flavors produced by the degradation of carotenoids and have a low olfactory threshold. Thus, small amounts of norisoprenoid compounds can have a significant organoleptic effect on the taste of food [11]. Therefore, the aroma quality of MJ can be improved if the degradation of carotenoids is fully utilized to increase the aroma of MJ [12].

Probiotic co-fermentation enhances flavor and increases the variety of fermented products. However, the use of non-*Saccharomyces* + LAB for the fermentation of MJ has not received much attention. This study aims to investigate the effect of mixed culture co-inoculation fermentations by using two non-*Saccharomyces* yeasts with two LAB on the physiochemical indices, TPC, antioxidant capacity, and volatile aroma and focuses on the carotenoid degradation related to the production of norisoprenoid compounds of fermented MJ. The outcome of this study may help develop different styles of FMJ.

## 2. Materials and Methods

### 2.1. Experimental Materials

Mature mango samples (variety: 'Hong mang' from a local mango farm in Changjiang, Hainan, China) used in this study were purchased from a supermarket in Haikou, Hainan, China. All fruits were free from evident mechanical damage.

MP (GDMCC140157), *Rhodotorula glutinis* (RG, GDMCC2.27), LP (GDMCC1.140), and *Lactobacillus casei* (LC, GDMCC1.159) were purchased from Guangdong Microbial Strain Conservation Center. YPD liquid medium and MRS broth were obtained from Hope Biotechnology Co., Ltd. (Qingdao, China).

### 2.2. MJ Fermentation

MJ was obtained by hand peeling, slicing, and squeezing the fresh and mature mango 'Hong' into a juice by using a household juicer (JYZ-E18; Joyoung Co., Ltd., Hangzhou, China) and mixed well with 50% (*v/v*) sterile water. The soluble solids of mango juice were adjusted to around 18 °Brix by adding sucrose and sterilized at 88 °C for 15 min. Activated LAB (5%, *v/v*) and non-*Saccharomyces* (5%, *v/v*) were inoculated into MJ at the same time. Fermentation was carried out at 28 °C for 72 h.

#### 2.2.1. Physicochemical Analysis

The total soluble solids (°Brix) and pH were measured at indicated time points by using a handheld refractometer (ATAGO PAL-1, Tokyo, Japan) and a pH meter (FE20 laboratory pH meter), respectively.

The organic acid content was determined using the high-performance liquid chromatography described previously [13]. The ZORBAXSB-Aq column (250 mm × 4.6 mm, 5 μm) was attached to a photodiode array detector on the UV spectrum (210 nm). The column was eluted with a mobile phase containing $NH_4H_2PO_4$ (0.01 M, pH = 2.62): methanol at a ratio of 97:3 at 30 °C and flow rate of 0.8 mL/min. The peaks of each organic acid were determined by comparing the sample retention time with those of standards. For each acid, a standard curve was constructed using standards to determine the relationship between peak area and concentration.

The sugar concentration was determined by referring to the method of Georgelis et al. [14]. The chromatographic column Anthenanh $NH_2$-RP (4.6 mm × 250 mm, 5 μm) connected to a RID differential detector was used. The column was eluted with a mobile phase of acetonitrile and water (7:3) at 35 °C and flow rate of 1 mL/min. The peaks of each sugar were determined by comparing the retention time of the sample with that of the standard.

### 2.2.2. TPC and Antioxidant Activity Assays

TPC was determined using the Folin–Ciocalteu method modified by Martins et al. [15]. In brief, 0.5 mL of sample and 0.25 mL of Folin–Ciocalteu reagent were added into a test tube. The mixture was left in the dark for 5 min, added with 0.5 mL of 12% $Na_2CO_3$ solution and 3.75 mL of distilled water, and left in the dark for another 2 h. The absorbance of the mixture was measured at 765 nm. Gallic acid was used as standard, and results were expressed as gallic acid equivalents (GAE).

The DPPH radical scavenging ability was determined in accordance with the method described by Escudero-López et al. [16]. The absorbance of the DPPH solution was diluted to 1.2–1.3 with ethanol at 540 nm. About 3.8 mL of DPPH solution was mixed with 0.2 mL of sample, and the mixture was left in the dark for 45 min. The absorbance of the mixture was measured at 540 nm. The DPPH radical scavenging capacity was calculated in accordance with the following equation:

$$DPPH (\%) = ([A \text{ control} - A \text{ sample}]/A \text{ control}) \times 100.$$

The 2,2′-azino-bis(3-ethylbenzothiazoline-6-sulfonic acid) (ABTS·) radical scavenging ability of FMJ was assayed using the method described by Thaipong et al. [17] with slight modifications. The ABTS solution (7 mM) was mixed with potassium persulfate solution (7.35 mM) and placed at room temperature, and protected from light for 16 h before use. About 0.1 mL of sample was added with 3.9 mL of ABTS radical solution and mixed well. Samples were placed in the dark for 10 min, and their absorbance was measured at 734 nm. The ABTS radical scavenging capacity was calculated using the following equation:

$$ABTS (\%) = ([A \text{ control} - A \text{ sample}]/A \text{ control}) \times 100.$$

The ferric ion-reducing ability was determined in accordance with the method described by Vadivel et al. [18]. The FRAP reagent was prepared using acetate buffer solution (0.3 M, pH 3.6), $FeCl_3·6H_2O$ solution (20 mM), and TPTZ solution (10 mM) at a ratio of 10:1:1 (*v/v/v*). The FRAP reagent (3.0 mL) was added to the sample (1.0 mL) and mixed well. The mixture was allowed to stand in the dark for 50 min. The absorbance of the mixture was measured at 593 nm. The standard curve was constructed using $FeSO_4$ as standard, and results were expressed as $FeSO_4$ equivalents.

The copper ion-reducing ability was determined by referring to the method described by Jin et al. [19]. About 1.0 mL of extracted sample, 1.0 mL of copper sulfate (10 mM), 1.0 mL of neocuproine solution (7.5 mM), and 1.0 mL of ammonium acetate solution (1 M) were mixed well. The absorbance of the sample at 450 nm was measured. Water-soluble vitamin E (Trolox) was used as a standard, and results were expressed as mM Trolox equivalents.

### 2.2.3. Carotenoids Degradation Rate Determination

The UV-visible spectra of MJ in the range of 350–600 nm were obtained using the full-wavelength scanning method. Considering the positive correlation between the mass concentration of carotenoids and the OD450 nm value, the OD450 nm value of the liquid culture was measured before and after inoculation.

The degradation of carotenoids in MJ was determined in accordance with the method of Ordóñez-Santos et al. [20]. About 1 g of MJ was weighed precisely and added with 4 g of tetrahydrofuran ethanol solution (10%, *v/v*). The mixture was shaken thoroughly, extracted by ultrasonication for 30 min, and centrifuged at 8000 r/min and 10 °C for 10 min. The supernatant was collected for the determination of OD450 nm.

Carotenoid degradation rate (%) = $([A_0 - A_1]/A_0) \times 100\%$, where $A_0$ is the light absorption value of the control at 0 h, and $A_1$ is the light absorption value of the sample after fermentation.

### 2.2.4. Aroma Component Analysis

Aroma components were extracted using headspace solid-phase microextraction and analyzed by gas chromatography–mass spectrometry (Agilent 7890A-5975C). The quadrupole was in the scanning mode (scanning range, 35–550 da; temperature, 150 °C; and electron ionization, 70 eV). The gas chromatographic column HP-5MS (60 m × 250 μm × 0.25 μm) was used to separate the aromatic compounds. The column temperature was initially set at 50 °C and held for 4 min and then increased to 100 °C for 2 min at a rate of 5 °C/min, 140 °C for 1 min at a rate of 4 °C/min, 180 °C for 2 min at a rate of 4 °C/min, and 250 °C for 5 min at a rate of 5 °C/min. Helium was delivered as carrier gas at a flow rate of 1.5 mL/min. The inlet temperature was 250 °C [21].

### *2.3. Data Analysis*

Experimental data were compiled using the Excel 16 software, and data were processed and analyzed by one-way analysis of variance by using the SPSS 24.0 software (SPSS Inc., Chicago, IL, USA). Results were expressed as mean ± standard deviation by using the Origin 2019b (OriginLab Co., Northampton, MA, USA). A heatmap was established using the R software (The University of Auckland, Auckland, New Zealand). The principal component was analyzed using Excel 16 and Xlstat software.

## 3. Results and Discussion

### *3.1. Changes in pH, Soluble Solid Content, and Organic Acid Content*

As shown in Table 1, the pH, °Brix, and organic acid contents of the LP + RG, LC + RG, LP + MP, and LC + MP groups were significantly different compared with those of the CK group after 72 h of fermentation ($p < 0.05$). The pH values of the LP + RG, LC + RG, LP + MP, and LC + MP groups decreased from 4.79 to 3.26, 3.33, 3.19, and 3.36, respectively. The pH of MJ co-cultured with LP decreased rapidly from 4.79 to 3.26 and 3.19. This finding might be related to the high lactic acid production during fermentation by using LP [22]. The total soluble solid contents of LP + RG, LC + RG, LP + MP, and LC + MP groups decreased from 18.37 °Brix to 14.67 °Brix, 14.80 °Brix, 13.83 °Brix, and 14.17 °Brix, respectively, and LP + MP showed strong sugar utilization ability.

Lactic acid is the main metabolite produced during the fermentation of LAB. As the number of LAB increases, the lactic acid content also increases [19]. Lactic acid was not detected in unfermented MJ. After 72 h of fermentation, the lactic acid concentration of co-cultured fermented MJ increased significantly ($p < 0.05$). The lactic acid contents of the LP + RG, LC + RG, LP + MP, and LC + MP groups increased to 7.77, 5.28, 6.10, and 4.20 g/L, respectively, accounting for about 50% of the organic acid content of each FMJ. This result indicated that lactic acid was the main organic acid metabolite formed during the fermentation of MJ.

**Table 1.** pH, soluble solids, and organic acids analysis of mango juice.

| Index | CK | LP + RG | LP + MP | LC + RG | LC + MP |
|---|---|---|---|---|---|
| pH | 4.79 ± 0.03 [a] | 3.26 ± 0.01 [c] | 3.19 ± 0.01 [d] | 3.33 ± 0.01 [b] | 3.36 ± 0.01 [b] |
| °Brix | 18.37 ± 0.06 [a] | 14.67 ± 0.06 [b] | 13.83 ± 0.06 [d] | 14.80 ± 0.10 [b] | 14.17 ± 0.05 [c] |
| Organic acid (g/L) | | | | | |
| Lactic acid | - | 7.77 ± 0.52 [a] | 6.10 ± 1.23 [b] | 5.28 ± 0.44 [c] | 4.20 ± 0.3 [d] |
| Malic acid | 5.97 ± 0.22 [a] | 1.81 ± 0.12 [c] | 2.21 ± 0.05 [c] | 5.30 ± 0.44 [b] | 5.31 ± 0.42 [b] |
| Citric Acid | 3.74 ± 0.07 [a] | 1.45 ± 0.14 [c] | 0.69 ± 0.02 [d] | 2.01 ± 0.09 [b] | 1.38 ± 0.05 [c] |
| Acetic acid | 1.81 ± 0.07 [a] | 0.24 ± 0.01 [e] | 0.59 ± 0.02 [d] | 1.36 ± 0.04 [b] | 1.27 ± 0.05 [c] |
| Tartaric acid | 1.50 ± 0.05 [b] | 1.09 ± 0.04 [c] | 2.04 ± 0.32 [a] | 0.66 ± 0.05 [d] | 1.82 ± 0.05 [a] |
| Oxalic acid | 0.56 ± 0.01 [a] | 0.18 ± 0.05 [c] | 0.31 ± 0.02 [b] | 0.12 ± 0.03 [d] | 0.19 ± 0.04 [c] |
| α-Ketoglutaric acid | 0.39 ± 0.01 [a] | 0.30 ± 0.01 [c] | 0.15 ± 0.02 [d] | 0.32 ± 0.01 [b] | 0.13 ± 0.01 [d] |
| Total | 13.97 | 12.84 | 12.09 | 15.05 | 14.30 |

The data of each sample are expressed as mean ± standard deviation ($n = 3$). Different letters ([a–e]) in the same row indicate significant differences at $p < 0.05$.

Malic and citric acids are the main organic acids in MJ and account for 38.15% and 23.90%, respectively, of the total organic acid content. High malic acid concentration can have a negative effect on the sensory properties of beverages [23]. When malic acid content is too high, MJ tastes sour. The use of LAB can convert malic acid into a less coarse and softer lactic acid [24]. An indicator of the fermentation process is the reduction rate of malic acid and the conversion of malic acid into lactic acid. For LP + RG, the highest decrease in malic acid content was observed after 72 h fermentation, showing the strongest reduction in malic acid [25]. The citric acid contents of FMJ in LP + RG, LC + RG, LP + MP, and LC + MP groups decreased significantly ($p < 0.05$) from 3.74 g/L to 1.45, 2.01, 0.69, and 1.38 g/L, respectively. LP + MP had the strongest citric acid utilization of 81.55%, and organic acid was the second carbon source for microbial reproduction during fermentation. Thus, this strain presumably uses citric acid as a carbon source to metabolize [26]. This metabolism in LAB and yeast co-cultures has also been described in other studies [19].

After 72 h of fermentation, the contents of oxalic and tartaric acids in MJ decreased. The reduction in tartaric acid concentration might be related to the formation of tartaric acid salt precipitates. During fermentation, these organic acids may interact with other substances, such as alcohols and aldehydes, to produce other flavor components [27].

Acetic acid was detected in unfermented MJ, but a significant decrease in acetic acid content occurred after 72 h of fermentation. The acetic acid concentrations of FMJ in LP + RG and LP + MP groups decreased from 1.81 g/L to 0.24 and 0.59 g/L, respectively. This result might be due to the consumption of acetic acid by LP as a carbon source. The decrease in acetic acid may be a positive factor because acetic acid may produce off-flavors at high concentrations [28].

*3.2. Changes in Sugar Content*

The consumptions of substrates (i.e., sucrose, fructose, and glucose) for 72 h of fermentation by LP + RG, LC + RG, LP + MP, and LC + MP are shown in Figure 1. The total sugar levels of FMJ in LP + RG and LC + RG groups decreased from 167.48 g/L to 97.78 and 104.85 g/L, respectively. Fructose, glucose, and sucrose levels were reduced from 26.68, 41.61, and 99.19 g/L, respectively, to 2.69, 0.47, and 70.36 g/L, respectively, by LP + MP and to 4.50, 1.88, and 77.79 g/L, respectively, by LC + MP. The sugar consumption of the different mixes revealed that the ability of MP to consume glucose and fructose was better than that of RG, and this finding was consistent with the results of Escribano et al. [29]. LP + MP/RG consumed glucose better than LC + MP/RG, and this finding might be because LP preferred glucose as a carbon source and could easily adapt to different conditions [30].

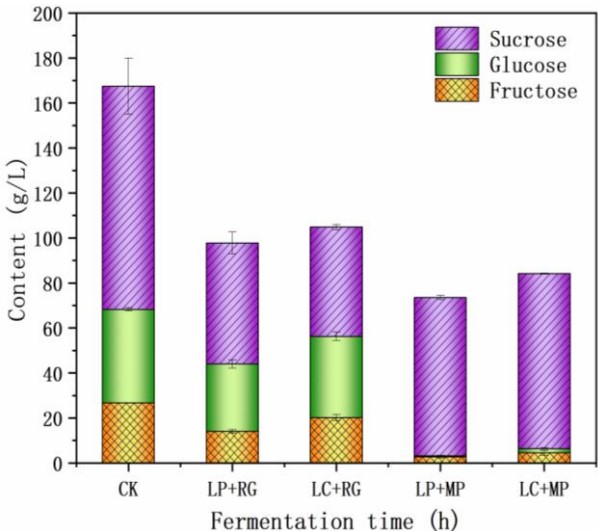

**Figure 1.** Sugar content changes in MJ before and after fermentation.

*3.3. Changes in Carotenoid Degradation Rate*

Peak absorption was found to be at 450 nm by scanning between 350 and 600 nm (Figure 2a). The magnitude of the absorbance value at 450 nm reflects the carotenoid content. A high absorbance indicates high carotenoid content and vice versa.

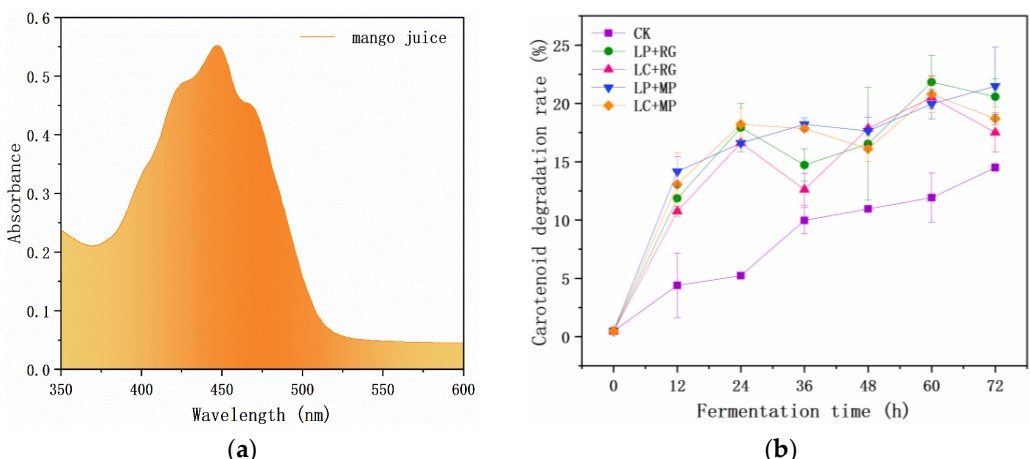

(**a**)                                                    (**b**)

**Figure 2.** (**a**) Full-wavelength scan of mango juice; (**b**) changes in the carotenoid degradation rate of mango juice before and after fermentation.

The changes in the degradation rate of carotenoids in MJ after fermentation are shown in Figure 2b. The degradation rates of carotenoids in MJ were 14.51% in the unfermented group and 20.57%, 17.50%, 21.48%, and 18.69% in the LP + RG, LC + RG, LP + MP, and LC + MP groups, respectively. Muntean et al. [31] evaluated the stability of carotenoids in courgette during lactic acid fermentation and observed different degradation rates of carotenoids due to exposure to acidic pH. Liu et al. [32] found that with increasing concentrations of lactic and acetic acids, pH and $\alpha$-carotene residuals decreased. Excess lactic acid results in the loss of $\alpha$-carotene, leading to the assumption that lactic acid accumulation has a facilitative effect on carotenoid degradation. Multari et al. [33] assessed the carotenoid composition of LAB-fermented 'Washington Navel' orange juice and similarly found that LAB fermentation results in a significant reduction in carotenoids in orange juice samples ($p < 0.05$). This result showed that this extensive loss could be due to the isomerization of molecules during fermentation and consequent oxidative degradation.

*3.4. Changes in TPC and Antioxidant Activity*

The changes in the TPC of FMJ are shown in Figure 3. At 0 h, the TPC of MJ (71.24 mg GAE/100 mL) was at its lowest. After 72 h of fermentation, the TPC of the LP + RG group was at its highest (123.92 mg GAE/100 mL), followed by the LP + MP (108.73 mg GAE/100 mL), LC + MP (87.47 mg GAE/100 mL), and LC + RG (84.01 mg GAE/100 mL) groups. Mashitoa et al. [30] and Hur et al. [34] noted an increase in TPC in lactic acid-fermented foods. In the present study, the TPCs in the LP + RG and LP + MP groups were higher than those in other groups. This finding might be due to the ability of LP to remove the sugar fraction and hydrolyze the glycosyl fraction of phenolic compounds during fermentation [35]. The increased TPC of fermented blueberry juice by LP is due to the hydrolysis of glycosides into sapogenins and the possible production of esterases that hydrolyze glycosidic ester bonds, which contribute to the release of phenolic compounds. Zhang et al. [36] and Minnaar et al. [37] investigated the effect of mixed culture co-inoculation on the phenolic and sensory characteristics of Syrah wines. A mixed culture co-inoculation strategy of Syrah wines with non-yeast, yeast, and LAB has resulted in improved phenolic and sensory characteristics compared with wines inoculated with brewer's yeast alone. The increase in TPC in FMJ might be attributed to the β-glucosidase produced by the organism during fermentation that increased the polyphenol content [38,39].

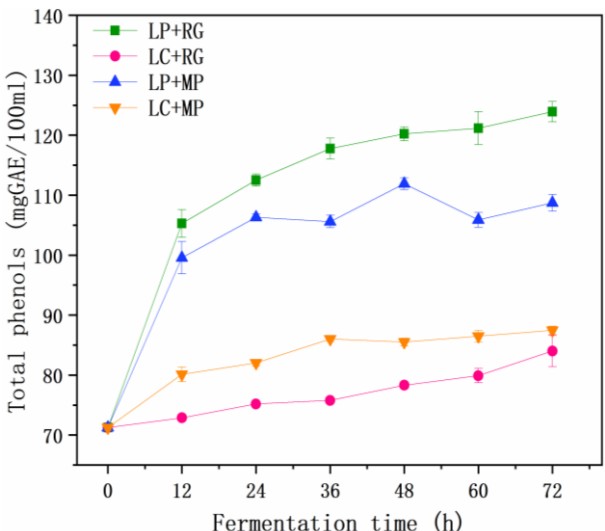

**Figure 3.** Total phenolic content (TPC) of mango juice before and after fermentation.

DPPH is a stable radical with a maximum absorbance of 540 nm in ethanol. When DPPH encounters a proton donor substance (e.g., antioxidant), the radical acts as a scavenger, and the absorbance decreases [40]. As shown in Figure 4A, an increasing trend in DPPH radical scavenging capacity was observed in all fermentation tests. After 72 h of fermentation, the DPPH radical scavenging capacities of the four groups of mixed FMJ, i.e., LP + RG, LC + RG, LP + MP, and LC + MP groups, increased significantly ($p < 0.05$) from 9.00% to 25.79%, 17.53%, 33.00%, and 19.55%, respectively. Jin et al. [9] reported that the DPPH radical scavenging ability of mango pulp decreases after 48 h of fermentation with LC. Mango pulp fermented with LP for 48 h had the highest scavenging capacity for DPPH radicals. Chen et al. [41] reported that different LAB-fermented kiwi pulp and LP obtained a higher scavenging rate than LC. This result was similar to our findings.

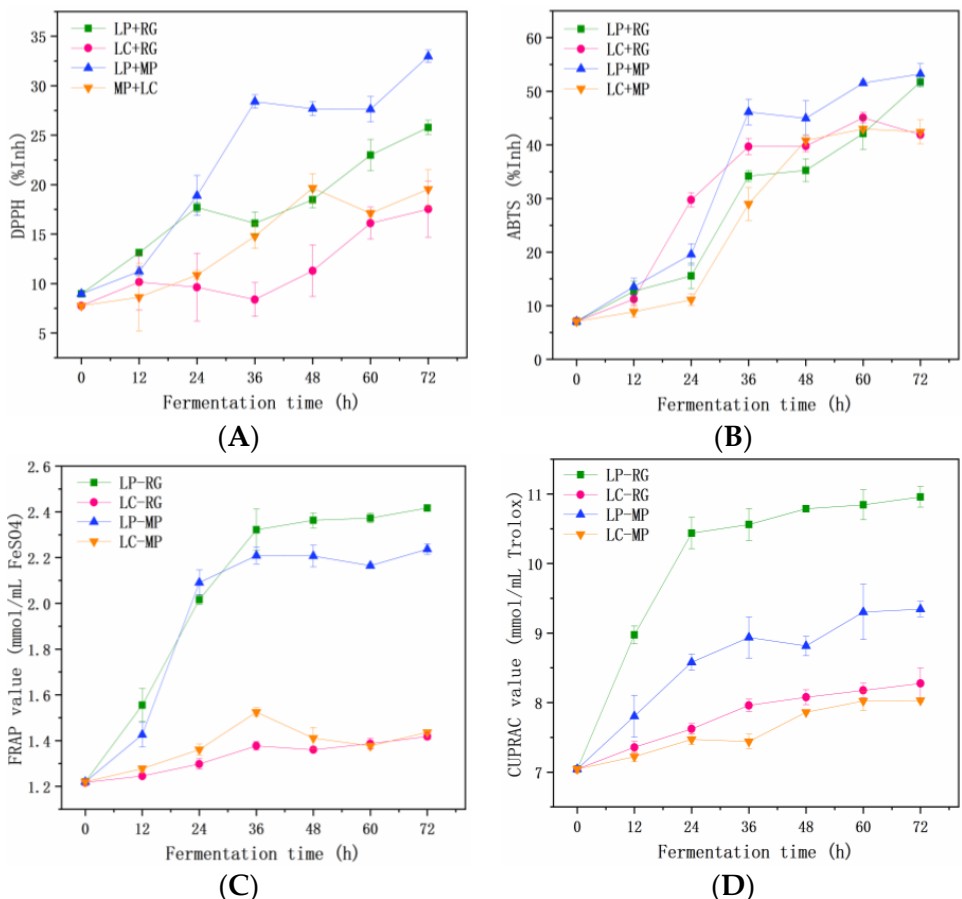

**Figure 4.** Changes in the antioxidant activities of fermented mango juice: (**A**) DPPH, (**B**) ABTS, (**C**) FRAP, and (**D**) CUPRAC assays. Data represent the mean ± standard deviation of each sample (*n* = 3). DPPH, DPPH radical scavenging activity; ABTS, ABTS radical scavenging activity; FRAP, ferric reducing antioxidant capacity; CUPRAC, cupric reducing antioxidant capacity; *L. plantarum*, *Lactobacillus plantarum*; and *L. casei*, *Lactobacillus casei*.

As shown in Figure 4B, all fermentation cases significantly increased the ABTS radical scavenging activity of MJ. After 72 h of fermentation, the ABTS radical scavenging capacities of MJ fermented with LP + RG, LC + RG, LP + MP, and LC + MP significantly ($p < 0.05$) increased from 7.09% to 51.75%, 41.91%, 53.23%, and 42.43%, respectively.

As shown in Figure 4C, FRAP values showed an increasing trend during the 72 h of fermentation. After 72 h of fermentation, the FRAP values of MJ fermented with LP + RG, LC + RG, LP + MP, and LC + MP increased from 1.12 mM $FeSO_4$ to 2.42, 1.42, 2.24, and 1.44 mM $FeSO_4$, respectively. At the end of fermentation, the FRAP values of MJ fermented with LP + RG exhibited stronger FRAP values than those of MJ fermented with LP + MP. The effect of LAB and yeast fermentation on the antimicrobial, antioxidant, and metabolomic properties of naturally carbonated probiotic whey beverages is investigated, and FRAP values in the experimental group are found to be significantly higher ($p < 0.05$) than those in the control whey [42]. The FRAP activity may be related to the phenolic, which is involved in the reduction of the 2,4,6 tris(2-pyridyl)-S-triazine (TPTZ)-$Fe^{3+}$ complex into the TPTZ-$Fe^{2+}$ form. The antioxidant activity of LAB is also associated with a number of structural polysaccharides and soluble secreted molecules [43].

As shown in Figure 4D, the CUPRAC values of MJ fermented with LC + RG and LC + MP increased slightly from 7.04 mM Trolox to 8.28 and 8.03 mM Trolox, respectively. After 72 h of fermentation, the CUPRAC values of MJ fermented with LP + RG and LP + MP increased significantly ($p < 0.05$) to 10.96 and 9.35 mM Trolox, respectively. These results indicated that the LP + RG group had the highest copper ion-reducing power.

### 3.5. Aroma Component Analysis

Table 2 shows that the basic major volatile components were roughly the same in MJ and FMJ. A total of 40 volatile components, including 11 alcohols, 3 esters, 7 alkenes, 9 ketones, 4 aldehydes, 5 acids, and 1 other, were observed.

Alcohols are the main products obtained by non-*Saccharomyces* yeast through amino acids during alcoholic fermentation. Eleven alcohols were identified in this study. 1-Hexanol, 1-heptanol, 1-octanol, phenylethyl alcohol, α-phellandren-8-ol, and 2-heptanol were found in all samples. The FMJ in LP + MP and LC + MP groups contained higher levels of phenylethyl alcohol than that in the CK group (Table 2). This study might be related to the ability of MP to produce a high concentration of phenylethyl alcohol from monosaccharides or phenylalanine [44].

Three esters, i.e., methyl syringate, ethyl decanoate, and diethyl phthalate, were identified in MJ. However, given that the odor detection thresholds for these compounds have not been determined, their contribution to the aroma of MJ is unclear. Studies pointed out that esters are predominantly related to juice composition, fermentation temperature, yeast strain, and degree of aeration [45].

Five acids were identified in MJ, with phenylacetic and octanoic acids being the most abundant. However, the contribution of these acids to flavor might be small as they usually have a high threshold for odor detection. Fatty acids are probably derived from the autoxidation of saturated lipids in the fruit [46].

Seven terpenoids were identified in five different treatments of MJ, including 3-carene (sweet, rosin flavor) [47], D-limonene (citrus-like and sweet taste) [40], linalool (citrus and floral notes) [48], caryophyllene (woody, green, spicy taste), and terpenic, humulene (green, herbaceous flavor) [48].

Carotenoids undergo chemical and enzymatic reactions to produce several compounds, such as TCH, beta-damascenone, and beta-ionone, some of which have powerful aromatic properties [6]. After 72 h of fermentation, the ketone contents (i.e., methylheptenone, geranylacetone, beta-damascenone, beta-ionone, and dihydro-beta-ionone) of FMJ in the LP + RG and LC + RG groups were significantly ($p < 0.05$) higher than that in the CK group, thereby supporting that the β-glucosidase secreted by RG had higher catalytic activity than *S. cerevisiae* in the conversion of aroma precursors [49]. These ketones provide FMJ with violet, floral, and fruity aromas and have an extremely low aroma threshold, thus making an important contribution to the aroma of MJ [50]. 2,5-Dimethyl-4-hydroxy-2H-furan-3-one (DMHF) is believed to be a key aroma constituent in many fruits and baked foods [51,52]. Studies pointed out that DMHF is an important aroma component in strawberries [53]. In the present study, given its odor activity value (OAV) > 1, DMHF is also important in the aroma of MJ.

Table 2 also lists the OAVs of several volatile compounds in FMJ. The significant contribution of each odor to the characteristic flavor can be determined by the OAV, which is the ratio of the concentration of the compound to the odor threshold. If a compound has an OAV > 1, it has a significant effect on the overall fruit odor [54]. If we consider the threshold concentration of a compound as a separate quantity, the OAV of a compound gives the amount of the threshold concentration of that compound in the fruit. A high OAV indicates a high probability of detection of the compound's odor. Of the 40 compounds evaluated, 13 compounds had levels above their threshold concentration (OAV > 1).

The hierarchical cluster analysis was used as a preliminary method to assess whether the volatile compounds associated with each sample could be clustered to the sample on the basis of the Euclidean distance drive of the sample [55]. This heatmap gave an overview of the variation of aroma compounds in MJ from four different mixed-probiotic fermentation treatments (Figure 5). Samples were clearly clustered into different places, and aroma compounds were clustered in accordance with different fermentation methods.

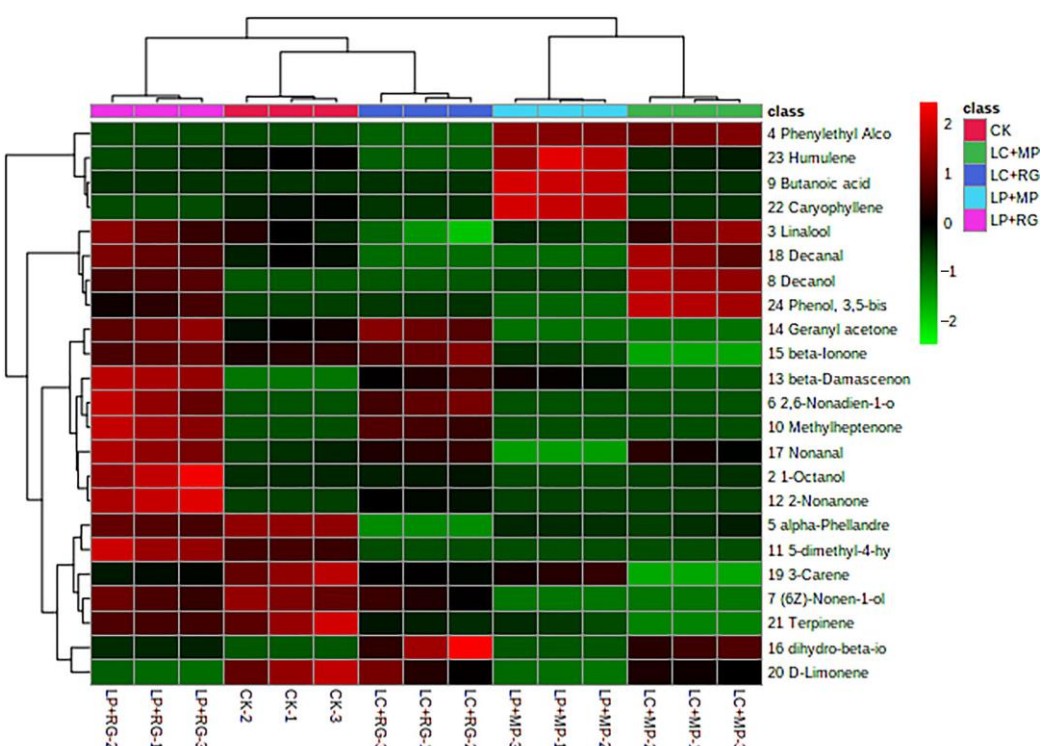

**Figure 5.** Heatmap analysis of mango juice before and after fermentation.

The first group was the aroma compounds produced in the LP + RG-fermented juices and included ketones giving balsamic, rose, and violet aromas. The second group was unfermented MJ containing terpene aroma compounds. The third category was the juice fermented by LC + RG, which had dihydro-beta-ionone with a woody and floral flavor. The fourth cluster of the thermogram represented the volatile compounds formed in FMJ in the LP + MP group, which contained a high proportion of phenylethyl alcohol, butanoic acid, caryophyllene, and humulene that had a woody and green flavor. The fifth cluster was characterized by including a high proportion of aldehydes with sweet, fatty, herbal, floral, and fruity notes.

### 3.6. Principal Component Analysis (PCA)

PCA is a multivariate data analysis technique used to downscale and show correlations between variables and samples [56]. This analytical method has been widely used in the aroma studies of fruit juices and fruit wines. This study used 24 aromatic compounds with OAV > 0.1 and 5 fermentation patterns as variables and analyzed the traits of each group by using the statistical method described above to show the aroma characteristics of MJ in a simple and effective way. About 27.31% and 71.64% of the total variance were accounted to PC1 and PC2, respectively. The sum of the two principal components was 98.95%, suggesting that these factors were sufficient for further discussion. The loadings of the aroma components of FMJ and the distribution of different fermentation treatments are shown in Figure 6.

**Table 2.** Main aroma components and content of fermented mango juice.

| Aroma Compound | LRI | Odour Threshold (µg/L) | Fragrance Description | CAS | OAV | Aroma substance Content µg/L | | | | |
|---|---|---|---|---|---|---|---|---|---|---|
| | | | | | | CK | RG + LP | RG + LC | MP + LP | MP + LC |
| Alcohol | | | | | | | | | | |
| 1-Hexanol | 860 | 600 [54] | Resin, flower, green | 111-27-3 | >0.1 | 62.43 ± 9.81 [a] | 46.92 ± 3.13 [b] | 17.63 ± 3.51 [c] | 7.79 ± 0.24 [d] | 2.81 ± 0.25 [d] |
| Citronellol | 1765 | 40 [56] | Green lemon | 106-22-9 | <0.1 | 2.05 ± 0.32 [a] | 2.13 ± 0.25 [a] | 1.81 ± 0.22 [a] | ND | ND |
| 1-Heptanol | 943 | 425 [54] | Fruity, herbaceous | 111-70-6 | <0.1 | 38.94 ± 4.91 [a] | 29.92 ± 3.00 [ab] | 23.42 ± 5.99 [bc] | 18.18 ± 1.89 [cd] | 6.18 ± 0.18 [d] |
| 3-Ethyl-3-pentanol | 1347 | Nf | Nf | 597-49-9 | Nf | ND | ND | ND | 12.77 ± 0.62 [a] | 6.40 ± 0.65 [b] |
| 1-Octanol | 1059 | 200 [54] | Honey, green, fatty | 111-87-5 | >0.1 | 28.44 ± 2.52 [bc] | 179.05 ± 27.38 [a] | 38.14 ± 3.50 [b] | 8.01 ± 0.44 [d] | 19.10 ± 4.76 [bc] |
| Phenylethyl Alcohol | 1136 | 1100 [56] | Floral1, rose, honey | 1960-12-8 | >0.1 | 242.64 ± 11.27 [c] | 247.30 ± 9.36 [c] | 88.37 ± 9.11 [d] | 1636.20 ± 55.54 [a] | 1515.37 ± 85.81 [b] |
| α-Phellandren-8-ol | 1572 | 4.6 [41] | Fresh, phenolic, woody | 1686-20-0 | >1 | 47.16 ± 0.05 [a] | 38.26 ± 2.23 [b] | 3.26 ± 0.17 [d] | 18.75 ± 0.61 [c] | 17.77 ± 2.58 [c] |
| 2,6-Nonadien-1-ol | 1152 | 4.5 [47] | Cucumber | 7786-44-9 | >1 | ND | 30.66 ± 5.43 [a] | 24.46 ± 2.98 [a] | ND | ND |
| (Z)-6-Nonenol | 1168 | 130 [54] | Melon, wax, green, and fat | 35854-86-5 | >0.1 | 37.25 ± 3.56 [a] | 30.82 ± 3.73 [b] | 24.30 ± 3.74 [c] | ND | ND |
| 2-Heptanol, 6-methyl- | 1356 | Nf | Waxy, fatty, and citrus | 4730-22-7 | Nf | 4.86 ± 0.15 [a] | 0.88 ± 0.15 [bc] | 0.19 ± 0.03 [c] | 5.93 ± 0.44 [a] | 1.15 ± 0.12 [b] |
| Decanol | 1208 | 40 [41] | Cucumber | 1120-06-5 | >0.1 | ND | 3.62 ± 0.16 [b] | ND | 0.55 ± 0.00 [c] | 5.34 ± 0.36 [a] |
| Acids | | | | | | | | | | |
| Hydrocinnamic acid | 1304 | 5000 [57] | Balsamic | 501-52-0 | <0.1 | 12.90 ± 1.92 [b] | 88.26 ± 1.90 [ab] | 23.63 ± 2.61 [b] | 102.13 ± 7.61 [a] | 27.72 ± 2.52 [b] |
| Butanoic acid | 775 | 24 [57] | Sweaty, rancid, yogurt | 65-85-0 | >0.1 | ND | ND | ND | 21.11 ± 0.94 [a] | 0.06 ± 0.00 [b] |
| Phenylacetic acid | 1248 | 2650 [16] | Sweet honey flavor c | 103-82-2 | <0.1 | 47.56 ± 6.58 [c] | 51.36 ± 5.90 [c] | 43.61 ± 3.62 [c] | 638.70 ± 38.41 [a] | 474.16 ± 3.98 [b] |
| Octanoic acid | 1173 | 3000 [54] | Rancid, cheese, fatty, sweat | 124-07-2 | <0.1 | 193.37 ± 31.21 [a] | 161.19 ± 22.28 [b] | 18.07 ± 3.67 [c] | 0.37 ± 0.07 [d] | 18.95 ± 5.40 [c] |
| Decanoic acid | 1372 | 10000 [54] | Fatty, rancid | 334-48-5 | <0.1 | ND | 10.64 ± 3.32 [a] | ND | ND | 8.94 ± 0.18 [b] |
| Esters | | | | | | | | | | |
| Methyl syringate | 895 | Nf | Nf | 2198-23-4 | Nf | ND | 55.20 ± 3.98 [a] | 9.54 ± 0.36 [b] | ND | ND |
| Ethyl decanoate | 1381 | 200 [16] | Fruity, wine-like, pear | 110-38-3 | <0.1 | 11.65 ± 1.60 [c] | 9.13 ± 1.16 [cd] | 18.87 ± 3.03 [b] | 5.61 ± 0.39 [d] | 25.98 ± 2.53 [a] |
| Diethyl phthalate | 1765 | Nf | Nf | 84-66-2 | Nf | ND | 18.40 ± 1.72 [b] | 0.25 ± 0.01 [c] | 23.59 ± 3.69 [a] | 19.71 ± 2.14 [b] |
| Ketones | | | | | | | | | | |
| Methyl vinyl ketone | 693 | Nf | Nf | 78-94-4 | Nf | ND | ND | ND | 17.32 ± 2.28 [a] | 0.24 ± 0.04 [b] |
| Methylheptenone | 969 | 50.00 [47] | Citrus, musty, grassy | 110-93-0 | >0.1 | ND | 47.87 ± 5.20 [a] | 30.67 ± 2.49 [b] | ND | ND |
| 5-dimethyl-4-hydroxy-3(2 H)-furanone | 1063 | 5 [16] | Candy cotton | 4077-47-8 | >1 | 10.40 ± 0.37 [b] | 16.57 ± 1.87 [a] | ND | ND | ND |
| 2-Nonanone | 1423 | 41 [58] | Fruity | 821-55-6 | >1 | ND | 80.75 ± 7.58 [a] | 19.51 ± 2.99 [b] | ND | ND |
| 2-Heptanone | 1180 | 140 [59] | Cinnamon, sweet | 110-43-0 | <0.1 | ND | 6.33 ± 1.24 [a] | ND | 1.91 ± 0.13 [b] | ND |
| β-Damascenone | 1821 | 0.002 [56] | Tobacco, apple, flora | 23726-93-4 | >1 | ND | 62.54 ± 4.04 [a] | 33.26 ± 6.00 [b] | 23.37 ± 2.52 [c] | 7.62 ± 0.61 [d] |
| Geranyl acetone | 1854 | 60 [56] | Magnolia, green, fruit | 3879-26-3 | >0.1 | 5.63 ± 0.65 [b] | 10.23 ± 1.10 [a] | 9.91 ± 1.06 [a] | ND | ND |
| β-Ionone | 1461 | 0.007 [56] | Balsamic, rose, violet | 79-77-6 | >1 | 45.25 ± 2.00 [b] | 55.60 ± 2.58 [a] | 57.87 ± 5.80 [a] | 25.21 ± 2.53 [c] | ND |
| dihydro-β-ionone | 1476 | 11 | Woody, floral | 31499-72-6 | >0.1 | ND | 0.45 ± 0.03 [c] | 1.94 ± 0.82 [a] | ND | 1.22 ± 0.15 [b] |

**Table 2.** *Cont.*

| Aroma Compound | LRI | Odour Threshold (μg/L) | Fragrance Description | CAS | OAV | Aroma substance Content μg/L | | | | |
|---|---|---|---|---|---|---|---|---|---|---|
| | | | | | | CK | RG + LP | RG + LC | MP + LP | MP + LC |
| Aldehydes | | | | | | | | | | |
| Benzaldehyde | 1523 | 350 [56] | Caramel, fruity, green | 100-52-7 | <0.1 | 7.47 ± 0.10 [b] | 0.87 ± 0.07 [c] | 28.31 ± 0.72 a | ND | ND |
| Nonanal | 1395 | 50 [56] | Rose-orange | 124-19-6 | >1 | 50.73 ± 5.55 [c] | 131.56 ± 10.98 [a] | 87.11 ± 4.69 [b] | 6.00 ± 0.22 [d] | 78.39 ± 8.98 [b] |
| Decanal | 1202 | 6 [54] | Fruity, citrus, orange | 112-31-2 | >1 | 5.89 ± 0.94 [c] | 11.87 ± 1.30 [b] | ND | ND | 13.59 ± 2.06 [a] |
| 2,4-dimethylbenzaldehyde | 1522 | Nf | Sweet, chemical | 15764-16-6 | Nf | 271.55 ± 16.40 [d] | 1767.29 ± 235.61 [b] | 2222.02 ± 177.03 [a] | 351.46 ± 24.71 [d] | 609.34 ± 31.53 [c] |
| Alkenes | | | | | | | | | | |
| 3-Carene | 948 | 50 [47] | Sweet, rosin | 13466-78-9 | >1 | 2533.56 ± 299.82 [a] | 1429.15 ± 73.20 [d] | 1514.35 ± 40.96 [cd] | 1768.46 ± 108.07 [bc] | 350.72 ± 11.72 [e] |
| D-Limonene | 108 | 10 [56] | Citrus-like, sweet | 5989-27-5 | >1 | 156.13 ± 22.27 [a] | 35.06 ± 3.55 [c] | 111.24 ± 28.64 [b] | 31.30 ± 2.64 [c] | 91.64 ± 6.50 [b] |
| Terpinene | 998 | 1000 [56] | weak citrus-like, fuel-like, dill, terpenic | 29050-33-7 | >0.1 | 105.65 ± 20.12 [a] | 78.17 ± 2.42 [b] | 43.52 ± 3.91 [c] | 34.34 ± 5.63 [c] | 9.70 ± 0.32 [d] |
| Linalool | 1097 | 3 [54] | Citrus, floral. Sweet, grape-like | 78-70-6 | >1 | 221.28 ± 19.89 [b] | 272.73 ± 22.56 [a] | 142.35 ± 25.04 [c] | 191.91 ± 10.10 [b] | 278.23 ± 29.11 [a] |
| Germacrene D | 1493 | Nf | Nf | 23986-74-5 | Nf | 0.22 ± 0.01 [b] | ND | ND | 5.82 ± 0.46 a | ND |
| Caryophyllene | 1494 | 64 [48] | Woody, green, spicy, terpenic | 87-44-5 | >1 | 130.06 ± 18.00 [b] | 59.24 ± 1.75 [d] | 91.95 ± 5.29 [c] | 396.95 ± 16.61 [a] | 84.87 ± 4.58 [c] |
| Humulene | 1645 | 120 [48] | Green, herbaceous | 6753-98-6 | >1 | 86.39 ± 9.16 [b] | 43.48 ± 10.52 [d] | 19.83 ± 2.47 [e] | 236.55 ± 28.44 [a] | 62.68 ± 6.95 [c] |
| Others | | | | | | | | | | |
| Phenol, 3,5-bis(1,1-dimethylethyl)- | 2315 | 200 [58] | Stone carbonate | 1138-52-9 | >0.1 | 22.30 ± 1.64 [cd] | 83.39 ± 15.92 [b] | 28.61 ± 2.02 [c] | 1.15 ± 0.03 [d] | 160.03 ± 8.51 [a] |

(1) Data for each sample are expressed as mean ± standard deviation (*n* = 3). Different letters ([a–e]) in the same row indicate significant differences at *p* < 0.05. (2) ND, Not detected. Nf. Not retrieved. (3) Odor thresholds and descriptors referenced from the literature.

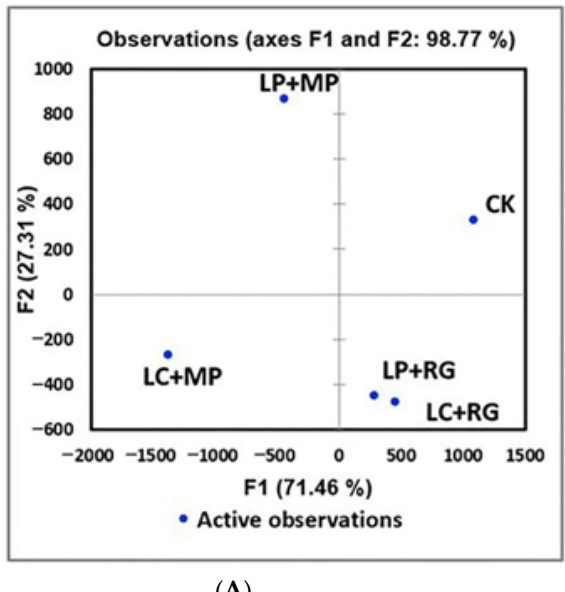
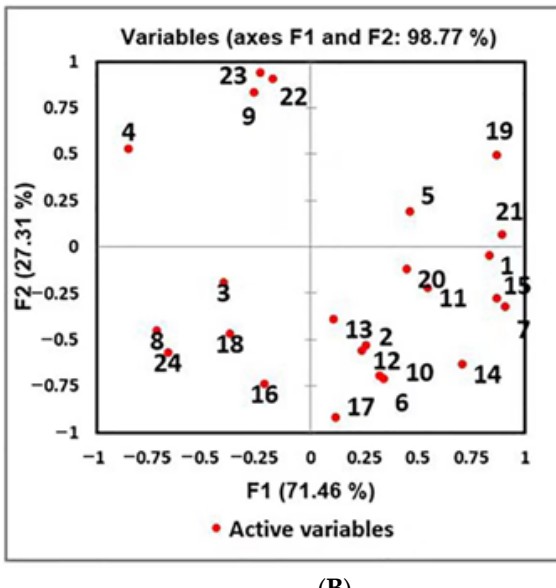

**(A)** **(B)**

**Figure 6. (A)** Dispersion diagrams of different microbial fermentation treatments of mango juice. **(B)** PCA plots of different aroma components: (1) 1-hexanol, (2) 1-octanol, (3) linalool, (4) phenylethyl alcohol, (5) alpha-phellandren-8-ol, (6) 2,6-nonadien-1-ol, (7) (6Z)-nonen-1-ol, (8) decanol, (9) butanoic acid, (10) methylheptenone, (11) 2,5-dimethyl-4-hydroxy-2 H-furan-3-one, (12) 2-nonanone, (13) beta-damascenone, (14) geranyl acetone, (15) beta-ionone, (16) dihydro-beta-ionone, (17) nonanal, (18) decanal, (19) 3-carene, (20) D-limonene, (21) terpinene, (22) caryophyllene, (23) humulene, and (24) 3,5-bis(1,1-dimethylethyl)-phenol.

Except for the alpha-phellandren-8-ol, 3-carene, and terpinene, volatile components were predominantly located in the second, third, and fourth quadrants. Unfermented MJ was located in the first quadrant, and most of the volatiles were located in the four quadrants near the negative part of PC2, close to the samples (LP + RG and LC + RG). They were closely related to 2-nonanone, beta-damascenone, and geranyl acetone. Thus, the co-fermentation of Lactobacillus with RG increased the content of ketones and aldehydes, whereas the co-fermentation of Lactobacillus with MP increased the contents of alcohols and terpenes.

*3.7. Norisoprenoid Analysis*

Volatile compounds are the main drivers of fruit juice and wine aromas. Carotenoid-derived volatiles with floral, fruity, fatty, and aromatic aromas are preferred by consumers. Given their low perceptual threshold, norisoprenoids may play an important role in juice and wine aromas [60]. Norisoprenoids are derived from the breakdown of carotenoids by enzymatic and acid-catalyzed hydrolyses. This breakdown leads to the formation of volatile aromatic compounds or nonvolatile sugar complexes in their free form within the juice [50,55]. Studies showed that the degradation of lycopene, α-carotene, and β-carotene produces methylheptenone, geranylacetone, beta-ionone, and alpha-ionone, thereby increasing the floral, fruity, fatty, and sweet aroma of the tomato fruit [58]. In the last two decades, Kaiser et al. [61] reported that the carotenoid-derived aromatic compounds beta-ionone and dihydro-beta-ionone have gained enormous importance in perfumery due to their floral intensity. Studies showed that beta-damascenone, which has tobacco, apple, and floral aromas, may act as a flavor-enhancing compound.

Significant differences in the amount of norisoprene aroma compounds were detected in all FMJ. For the MJ, LP + RG, LC + RG, LP + MP, and LC + MP groups, the norisoprene aroma compound contents were 50.88, 176.69, 133.65, 48.58, and 8.84 μg/L, respectively, after 72 h of fermentation ($p < 0.05$) (Figure 7). A high amount of norisoprene aroma compounds produced by the MJ fermented with RG was attributed to the high level of beta-

glucosidase hydrolysis capacity. Hu et al. [62] reported that the glycosidase extract of RG has improved catalytic selectivity for the 'fruity and floral' glycosides of C13-norisoprene compounds. Martínez et al. [63] proved that the yeast RG, which produces β-glucosidase, is naturally present during the fermentation process. The mixed fermentation of *R. mucilaginosa* and *S. cerevisiae* improves the composition of aromatic compounds in the variety and during fermentation, enhancing the aromatic characteristics of citrus, sweet, and sour fruits, berries, and flowers [49].

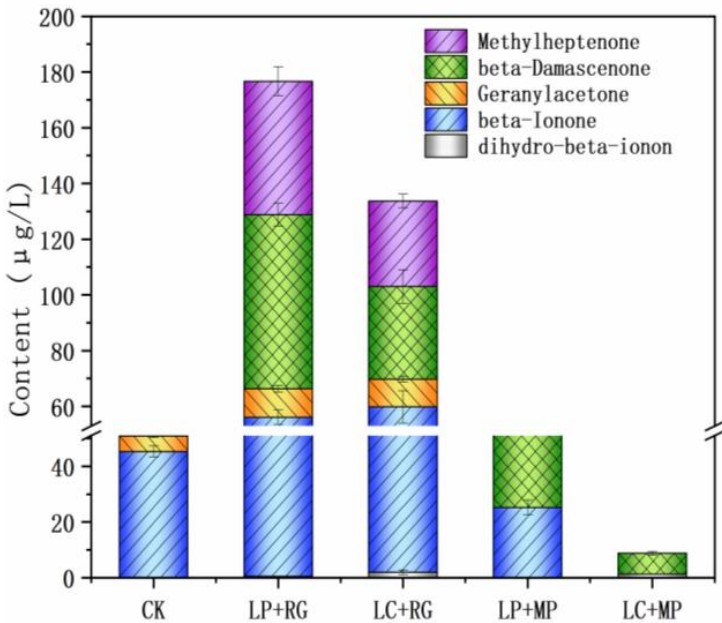

**Figure 7.** Changes in norisoprenoids before and after fermentation of mango juice.

## 4. Conclusions

This study assessed the effects of four mixed culture co-inoculation combinations of LAB with non-*Saccharomyces* on the physicochemical properties, phenolic substances, antioxidant properties, carotenoid degradation, and flavor characteristics of MJ. All MJ were found to have varying degrees of carotenoid degradation, and the mixed culture co-inoculation strategy of LAB with non-*Saccharomyces* resulted in improved physicochemical and flavor attributes of MJ compared with those of unfermented MJ. The co-inoculation of mixed cultures of non-*Saccharomyces* and LAB provided a practical method for improving the quality of MJ. The co-inoculation of mixed cultures of RG and LP resulted in increased TPC and enhanced DPPH radical scavenging activity, ABTS radical scavenging activity, iron-reducing antioxidant capacity, and copper-reducing antioxidant capacity. The analysis of aroma components showed that the co-inoculation of RG and LP increased the content of aroma compounds. The increased content of norisoprene aroma compounds, such as methylheptenone, geranylacetone, beta-damascenone, beta-ionone, and dihydro-beta-ionone, enhanced the pleasant aroma characteristics of MJ, such as floral and fruity notes.

Results also suggested that the mixed culture co-inoculation was a viable strategy, but success depends on the consideration of appropriate combinations of non-Saccharomyces/ LAB. The interactions between RG and LP during fermentation and the mode of inoculation are complex and require further research.

**Author Contributions:** Conceptualization, Q.Z., W.C. (Weijun Chen); methodology, R.C.; formal analysis, R.C., M.Z., W.C. (Wenxue Chen), H.C. and Q.Z.; investigation, R.C.; resources, R.C.; data curation, R.C.; writing—original draft preparation, R.C.; writing—review and editing, R.C.; visualization, R.C.; supervision, Q.Z.; project administration, Q.Z.; funding acquisition, Q.Z. All authors have read and agreed to the published version of the manuscript.

**Funding:** This research was funded by the National Natural Science Foundation of China (No. 31960509).

**Institutional Review Board Statement:** Not applicable.

**Informed Consent Statement:** Not applicable.

**Data Availability Statement:** Data are contained within the article.

**Conflicts of Interest:** The authors declare no conflict of interest.

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
