# Peer review of "Effect of the Mixed Inoculation of Lactic Acid Bacteria and Non-Saccharomyces on the Quality and Flavor Enhancement of Fermented Mango Juice"

_fermentation, doi:10.3390/fermentation9060563_

Round 1
Reviewer 1 Report
The manuscript " Effect of the mixed inoculation of lactic acid bacteria and non - Saccharomyces on the quality and flavor enhancement of fermented mango juice" is well written, the idea of this paper is very interesting.
The title of the manuscript is suitable of researching. Aim of the paper is clear. Introduction, Materials and Methods and Results and Discussion are well written. The conclusion is connected with the obtained results and the contribution of the research is highlighted.
Manuscript needs minor improvement prior its publication.
The authors should calculate and add in Table 2. linear retention indices (LRI) of aroma compounds instead retention time.
Check spelling and grammar throughout the paper. The Tables are presented at a very low level of quality and need to be improved.
Results (Table 1. and Table 2.) should be presented with superscript letters (a, b, c…).
Author Response
Response to Reviewer 1 Comments
Point 1: The authors should calculate and add in Table 2. linear retention indices (LRI) of aroma compounds instead retention time.
Response 1: Thank you for your valuable advice. The linear retention indices (LRI) of aroma compounds were instead retention time in table 2.
Point 2: Check spelling and grammar throughout the paper. The Tables are presented at a very low level of quality and need to be improved.
Response 2: Thank you for your advice. And it has been revised as your suggestion.
Point 3: Results (Table 1. and Table 2.) should be presented with superscript letters (a, b, c…).
Response 3: Thank you for your advice. And it has been revised as your suggestion.

Reviewer 2 Report
The manuscript desribes the preparation of fermented mango juice using a non-Sacharomyces species and LAB. The work carried out is novel in nature and falls within the aims and scope of Fermentation journal. The work flow needs some improvement to meet the high quality standrads of the journal. For example, sensory analysis section and sensory analysis results is missing. The authors should provide in details the sensory analysis results. They can also run statistical analysis in these results.
A graphical abstract is also required. It gives the reader a graphical shot of the work carried out.
Some other comments that must be addressed are given within the attached pdf.
Based on the overall quality of the manuscript and data interpretation (is very good in general) I suggest a minor revision prior to further consideration for publication.

The English language is good, it needs moderate improvement in some cases. See attached file.
Author Response
Response to Reviewer 2 Comments
Point 1: The manuscript desribes the preparation of fermented mango juice using a non-Sacharomyces species and LAB. The work carried out is novel in nature and falls within the aims and scope of Fermentation journal. The work flow needs some improvement to meet the high quality standrads of the journal. For example, sensory analysis section and sensory analysis results is missing. The authors should provide in details the sensory analysis results. They can also run statistical analysis in these results.
Response 1: We all think well of your thoughtful suggestion, but the sensory analysis results will be arranged to publish in another paper.
Point 2: A graphical abstract is also required. It gives the reader a graphical shot of the work carried out.
Response 2: We all think well of your thoughtful suggestion.
Point 3: Some other comments that must be addressed are given within the attached pdf.
Response 3: Thank you for your advice. And it has been revised as your suggestion according to the attached pdf.
